# Sex Steroids and Brain-Derived Neurotrophic Factor Interactions in the Nervous System: A Comprehensive Review of Scientific Data

**DOI:** 10.3390/ijms26062532

**Published:** 2025-03-12

**Authors:** Gilmara Gomes de Assis, Maria Bernardete Cordeiro de Sousa, Eugenia Murawska-Ciałowicz

**Affiliations:** 1Escola Superior Desporto e Lazer, Instituto Politécnico de Viana do Castelo, Rua Escola Industrial e Comercial de Nun’Álvares, 4900-347 Viana do Castelo, Portugal; 2Sport Physical Activity and Health Research & Innovation Center, 4900-347 Viana do Castelo, Portugal; 3Brain Institute, Federal University of Rio Grande do Norte, Natal 59078-970, RN, Brazil; mbcsousa@neuro.ufrn.br; 4Department of Physiology and Biomechanics, Wroclaw University of Health and Sport Sciences, 51-612 Wrocław, Poland; eugenia.murawska-cialowicz@awf.wroc.pl

**Keywords:** estrogen, estradiol, testosterone, progesterone, brain-derived neurotrophic factor, nervous system

## Abstract

Sex steroids and the neurotrophin brain-derived neurotrophic factor (BDNF) participate in neural tissue formation, phenotypic differentiation, and neuroplasticity. These processes are essential for the health and maintenance of the central nervous system. Aim: The aim of our review is to elucidate the interaction mechanisms between BDNF and sex steroids in neuronal function. Method: A series of searches were performed using Mesh terms for androgen/receptors, estrogen/receptors, and BDNF/receptors, and a collection of the scientific data available on PubMed up to February 2025 about mechanical interactions between BDNF and sex steroids was included in this literature review. Discussion: This review discussed the influence of sex steroids on the formation and/or maintenance of neural circuits via different mechanisms, including the regulation of *BDNF* expression and signaling. Estrogens exert a time- and region-specific effect on BDNF synthesis. The nuclear estrogen receptor can directly regulate *BDNF* expression, independently of the presence of estrogen, in neuronal cells, whereas progesterone and testosterone upregulate *BDNF* expression via their specific nuclear receptors. In addition, testosterone has a positive effect on BDNF release by glial cells, which lack androgen receptors.

## 1. Introduction

Sex steroid hormones, i.e., androgen, estrogen, and progesterone, are well known for their role in stimulating the reproductive organs and developing secondary sexual characteristics in females and males [1]. These hormones share similar structures and can reach the brain across the blood–brain barrier when released from peripheral steroidogenic organs [2] or are capable of being synthesized de novo within the brain [3], whereby they regulate neuronal functions. The synthesis of neuroactive steroids occurs primarily in neurons, as well as in microglia and astrocytes, and requires the translocation of cholesterol across the mitochondrial membrane through a molecular complex formed by the translocator protein 18 kDa, the steroidogenic acute regulatory protein, the voltage-dependent anion channel protein, and the adenine nucleotide transporter protein. Cholesterol is converted to pregnenolone by enzymatic activity in the mitochondria, and pregnenolone diffuses into the cytosol, where it is further metabolized by different enzymes into different neuroactive steroids (see Figure 1) [4,5].

In addition to sexual dimorphism present in several neural circuits, sex steroids generally play a neuroprotective role and exhibit different impacts on the brain’s function [6]. Two main effects of steroid hormones on the brain are described to produce sexual dimorphism in vertebrates: the first is organizational, associated with changes during sensitive phases of development, such as prenatal and/or perinatal sexual differentiation and adolescence; and the second refers to activation effects associated with sexually dimorphic phenotypic changes during puberty and throughout adulthood [7]. Furthermore, experimental studies show that learning responses are influenced differently by sex steroids in males and females through various mechanisms underlying neuroplasticity, i.e., learning-induced and hormone-driven brain region- and cell type-specific cellular activity and morphology [8]. The exposure of undifferentiated neural stem cells to different sex steroids results in phenotypic changes with alterations in functional and behavioral responses [9,10].

The brain-derived neurotrophic factor (BDNF) is a conserved neurotrophin that participates in multiple processes related to brain development and functioning, i.e., neuronal cell migration and differentiation, axonal growth, and synaptogenesis. The actions of sex steroids overlap with those of BDNF in various neural plasticity processes; however, the mechanisms underlying such interactions are not fully elucidated [11,12,13,14]. Further, a possible time- and region-specificity for sex steroid effects on BDNF activity adds an extra complexity to this topic [15,16].

The presence of an estrogen-responsive element (ERE) in *BDNF* gene sequences started the investigations on the participation of sex hormones in the development and organization of the nervous system. The interaction between estrogen and BDNF was initially evidenced by the colocalization of BDNF and TrkB mRNAs in estrogen-sensitive neurons of the developing cerebral cortex [17]. Later, Singh, Meyer, and Simpkins [18] pointed out that estrogen was more effective in maintaining BDNF levels in the hippocampus than in the cortex of female rats. This supports the thesis that the steroid regulation of *BDNF* expression is influenced by neurosteroid availability.

Estrogen activation and translocation of the nuclear estrogen receptors (ERs) to the cell nuclei can directly regulate the expression of *BDNF* via its ERE sequence. This is called the ‘genomic mechanism’. Additionally, membrane-associated steroid receptors can activate extra nuclear-initiated kinase signaling and modify intracellular mechanisms, amplifying steroids’ effect on gene expression [19,20]. For instance, the estrogen activation of the G-protein coupled membrane ER activates the mitogen-activated protein kinase (MAPK) and phosphoinositide 3-kinase (PI3K) pathways. As the kinases downstream of the MAPK and protein kinase (PKA) pathways increase ER phosphorylation, this potentiates estrogen signaling [21].

BDNF signaling via tyrosine kinase (Trk) B receptors within neuron membranes various intracellular signaling pathways, including MAPK/extracellular signal-regulated protein kinase (ERK), phospholipase Cγ (PLCγ), and PI3K pathways that induce the synthesis of proteins that regulate neuronal survival and synaptic maintenance [22]. This implies that BDNF function in synaptic plasticity is partially shared with that of estrogen.

Despite many studies addressing estrogen’s influence on BDNF activity, the regulation of BDNF by androgen and progesterone is also seen to participate in neuroplasticity processes in females and males. Progesterone exerts its effects on BDNF through genomic and non-genomic mechanisms. Although the induction of *BDNF* expression requires nuclear PR, the release of BDNF can be mediated by activating a distinct membrane-associated PR. Additionally, some membrane-associated PRs may lead to the activation of ERK1/2, which may exert an inhibitory influence on *BDNF* expression [23].

Considering the above-mentioned findings, in this literature review, we collected experimental evidence available up to February 2025 on the interactions between sex steroids (estrogens, testosterone, and progesterone) and BDNF from studies in vitro, in vivo, and ex vivo and provided a qualitative synthesis of the mechanisms by which sex steroids participate in neuroplasticity via BDNF modulation.

## 2. Methods

To assess the scientific literature addressing mechanistic interactions between sex steroids and BDNF, a broad search was carried out on PubMed Central and National Institute of Health platforms. For that, various combinations of PubMed MeSH terms were applied to the advanced PubMed search as follows: Each androgen MeSH term (testosterone, androgen receptors, androgens, dehydroepiandrosterone, and dihydrotestosterone) was combined with each BDNF MeSH term, (brain-derived neurotrophic factors, TrkB receptor, and brain-derived neurotrophic factor precursor). Each estrogen or progesterone MeSH term (estrogen, estradiol, estrone, estriol, progesterone, estrogen receptor, estradiol receptor, estrone receptor, and estriol receptor) was combined with each BDNF MeSH term (brain-derived neurotrophic factors, TrkB receptor, and brain-derived neurotrophic factor precursor). After duplicate removal, the retrieved papers were screened by two independent researchers for inclusion criteria according to the PECOS strategy: P (Population—animal/cell/tissues used for analysis of changes in BDNF expression/concentrations produced by sex steroids), E (exposure—sex steroid hormone analyses or experimental treatments), C (comparison—placebo or control conditions) O (outcomes—BDNF/TrkB expression/protein analyses) S (study design—in vivo/in vitro/ex vivo studies). The last search update occurred on 10th February 2025. Clinical studies, reviews, experimental models of hormonal disease, and studies that do not address BDNF and sex steroids in the same sample, as well as those not presenting comparable groups or validated measurement tools, were excluded from the qualitative synthesis. Ninety-nine studies in vitro, in vivo, or ex vivo reporting data from the analysis of sex steroids and BDNF in the same cell/tissue were included in the qualitative synthesis (Figure 2).

## 3. Estrogen and BDNF Interactions

Sex steroid hormones regulate BDNF activity in many neuronal circuits and at different phases of the lifespan. They act on brain tissues at different stages of development, specifically during the perinatal sensitive period, as well as adolescence, to organize sexual differentiation and produce sex-specific behaviors throughout the lifetime [1]. The scientific data supporting sex steroid interactions with BDNF in neural tissues will be discussed in the following subsections, and brief summaries of our findings are displayed in Table 1, Table 2 and Table 3.

The conserved co-regulation of genes such as ER, BDNF, and TrkB through different species indicates that the genes have closely associated functions and are part of the same pathway [81,82]. These gene expression patterns in developmental estrogen target neurons were the first pieces of evidence indicating that BDNF expression is related to neuron sensitivity to estrogen [17]. In line with this, Singh et al. [18] showed that ovariectomy (OVX) results in a reduction in BDNF levels in the cortical tissue. Associations between reductions in ER and BDNF levels have been reported by several studies.

From another perspective, ER activation was seen to increase BDNF levels and synaptogenesis in the cerebellum of neonatal mice [55], the pre-frontal cortex of rats [73], and in zebrafish [74]. The regulatory role of ERs over BDNF results in positive cognitive outcomes [55,73,74]. Indeed, estrogen directly regulates the expression of BDNF in neural cells [28,34,37,39,42,48,51,54,59,60,69]. Additionally, ER can regulate BDNF expression and function via the transactivation of TrkB receptors [38,49,52,71,83,84]. In addition to EREs, the activation of TrkB receptors also potentiates estrogen’s effects via ERα phosphorylation downstream of the TrkB/MAPK/ERK pathway. Meanwhile, the activity of the PI3K/AKT pathway constitutively inhibits basal transcription at EREs and inhibits TrkB-dependent transcriptional activation at EREs [50]. This grants homeostatic equilibrium to estrogen regulation on BDNF function. For instance, the disruption of estrogen regulation in BDNF expression leads to an elevation in TrkB phosphorylation [53,64,75,78].

The functional interaction between estradiol and BDNF participates in activity-dependent dendritogenesis. Murphy, Cole, and Segal [25] detected a short-term decrease in BDNF concentrations in hippocampal neurons treated with estradiol, in which BDNF declined up to 24 h and recovered within 48 h. This affected the activity-dependent formation of dendritic spines in GABAergic hippocampal neurons.

Zhu et al. [56] investigated whether estradiol regulates BDNF expression in the hypothalamus during the estrous cycle in rats. BDNF expression transiently increases in the ventromedial nucleus of the hypothalamus following the estradiol peak. Balasubramanian et al. [61] added that a chronic exposition of the amygdala to estradiol leads to a decrease in BDNF expression levels and behavioral implications. Cavus and Duman [33] reported that BDNF levels decrease in the dentate gyrus and the medial prefrontal cortex when estradiol levels are highest during estrus. And that acute estradiol treatment decreased hippocampal BDNF expression in acute OVX rats, with no effect in chronically OVX rats.

These findings indicate that the effect of cell exposition to estradiol on BDNF expression is influenced by time and intensity and involves selective mechanisms according to the neuronal population.

There are at least three types of ERs: estrogen receptor alpha (ERα) and beta (ERβ), which are localized in both the nuclei and the cell membrane, and G-protein coupled ER1 (GPER1), which is found at the cell membrane [85]. ERα and ERβ have specific expression patterns and show different biological functions across tissues, potentially due to their unique sets of downstream target genes [86] or to structural changes. ERβ has a weaker transcriptional activation function, which is attributed to differences in the amino-terminal domains [87]. Recent studies showed that ERα, but not ERβ, participates in skeletal muscle plasticity and mitochondriogenesis via BDNF expression regulation [88,89].

Nuclear ERs (α and β) are present in cells from multiple brain regions, e.g., the hippocampus, hypothalamus, amygdala, and prelimbic cortex [31,32,36,44,45,66,70,77,90]. Recently, the authors of [80] demonstrated that the activation of GPER1 promotes BDNF/TrkB signaling in hippocampal cells. Furthermore, the activation of estrogen-related receptor γ regulates BDNF expression in dopaminergic neurons [76] and is seen to ameliorate depression-like behavior and enhance neurogenesis in the hippocampus via the upregulation of BDNF/TrkB signaling [79]. These findings lead to the assumption that the effects of cell exposition to estrogens on BDNF expression are also influenced by the type and expression patterns of ERs.

The study by Wu et al. [62] reported that neither OVX nor estradiol treatment altered *BDNF* expression in the hippocampus of *BDNF* Val66Met heterozygote mice. OVX reduced *BDNF* expression solely in the hippocampus of wild-type mice, and the estradiol-induced increase in BDNF was restored to the dorsal hippocampus of wild-type OVX mice. The control of BDNF function requires the enzymatic cleavage of the pro-apoptosis pro-BDNF isoforms into the pro- survival mature BDNF. The Val66Met polymorphism negatively affects *BDNF* expression and protein processing into secretory pathways, altering the released pro-BDNF/BDNF [91,92]. It is possible that Val66Met heterozygote mice already presented altered hippocampal levels of BDNF such that these levels were not substantially changed by OVX or estradiol treatments.

The enzymatic activity of aromatase plays a relevant role in the regulation of BDNF expression by estrogen [30,35]. Aromatase is a membrane-bound enzyme located in the endoplasmic reticulum in estrogen-producing cells, which catalyzes the desaturation (aromatization) of the ring A of C19 androgens and converts them to C18 estrogens. Dittrich et al. [93] demonstrated that estradiol-induced BDNF expression in the forebrain song control nucleus of male juvenile zebra finches is decreased by the selective aromatase inhibitor fadrozole.

In line with this, aging affects estrogen regulation on BDNF. Data from Jezierski and Sohrabji’s [26,27] studies report that estrogen increases the expression of TrkB receptors in the brains of young animals but not in older ones. The data also found that aging impairs estrogen regulation on pro-apoptotic p75 neurotrophic receptor expression, favoring its increase and influencing neuronal apoptosis. It is possible that the ERE in the BDNF gene becomes less sensitive to estrogen with age, and this leads to a loss in the BDNF regulation of TrkB receptors. Moreno-Piovano et al.’s [58] comparison of mice with short-term vs. long-term OVX concluded that the effect of estradiol on BDNF expression declines over time. Sex steroids receptors regulate a negative feedback in response to persistent stimulation via a decrease in transcriptional activity, ubiquitination or degradation of steroid receptors is order to keep homeostasis. Therefore, a possible drawback of ER expression to long-term exposure to estrogen might contribute to the loss of estrogen effect on BDNF regulation.

From a physiological perspective, exercise was shown to improve estradiol regulation on *BDNF* expression [29] and to restore hippocampal levels of BDNF in OVX animals. These preliminary studies report that rats with 14 days of free-wheel access, running on a treadmill at low intensity for 30 min for 20 days, or combined 3 days of resistance training and 3 days of running for 8 weeks displayed improvements in sex steroid regulation on *BDNF* expression in their hippocampi [40,65,72,94].

## 4. Progesterone and BDNF Regulation

Early experiments by Gibbs et al. [24,95] demonstrated that levels of BDNF fluctuate across the estrous cycle and increase in response to acute hormone replacement. Murphy et al. [25,96] found that estradiol transiently reduces BDNF and GABA synthesis in hippocampal cells and, consequently, reduces inhibitory GABAergic connections, leading to a brief enhancement in neuronal activity, after which BDNF returns to normal levels and the inhibitory tone is restored. They detected that progesterone was able to block the increase in CREB phosphorylation and prevent estradiol-induced increases in cell activity and spine density. In line with this, Aguirre et al. [97] showed that progesterone reverses the estradiol-induced increase in ER and BDNF levels and eliminates the estradiol effect against glutamatergic excitotoxicity in hippocampal cultures. It can be seen that progesterone inhibition on CREB activity has a negative influence on estradiol effects on BDNF in GABAergic neurons. Additionally, Franklin & Perrot-Sinal [41] reported that young adult female rats treated with estradiol have higher hippocampal BDNF levels than those treated with combinations of estradiol and progesterone.

**Table 2 ijms-26-02532-t002:** Data summary of interactions between progesterone and BDNF.

	Title	Design/Methods	Aim/Experiment	Main Findings
1	Treatment with estrogen and progesterone affects relative levels of brain-derived neurotrophic factor mRNA and protein in different regions of the adult rat brain Gibbs, 1999. [95]	Ex vivo, RT-PCR, ELISA.	To examine the acute effects of estrogen and PROG on levels of BDNF expression and protein in different brain regions in adult mice.	Estrogen or estrogen + PROG increased BDNF expression and protein levels in the pyriform cortex of mice. Increases in BDNF expression in the hippocampus accompanied a decrease in BDNF protein.
2	Progesterone prevents estrogen-induced dendritic spine formation in cultured hippocampal neurons Murphy and Segal, 2000. [96]	In vitro, Hippocampal cultures, Immunocyto chemistry.	To examine PROG effects on estrogen-induced formation of dendritic spines in hippocampal cell cultures.	PROG did not affect the estrogen-induced downregulation of BDNF, but it did block the effect of estrogen on CREB phosphorylation.
3	Progesterone counteracts estrogen-induced increases in neurotrophins in the aged female rat brain Bimonte-Nelson, Nelson, and Granholm, 2004. [98]	Ex vivo, ELISA.	To test estrogen and estrogen + PROG effects on neurotrophin levels in cognitive brain regions in aged OVX mice.	Estrogen treatment increased BDNF, NGF, and NT3 levels in the mice’s entorhinal cortexes, and PROG abated these effects, and dropped BDNF levels in aged OVX non-treated mice.
4	Progesterone up-regulates neuronal brain-derived neurotrophic factor expression in the injured spinal cord González et al., 2004. [99]	Ex vivo, In situ hybridization, Immunocytochemistry.	To demonstrate that BDNF increases with PROG treatment in ventral horn motoneurons from spinal cord injured mice.	Spinal cord injury reduces BDNF expression levels by 50% in spinal motoneurons. PROG enhances BDNF in the motoneurons of lesioned spinal cord mice.
5	Progesterone treatment of spinal cord injury: Effects on Receptors, Neurotrophins, and Myelination De Nicola et al., 2006. [100]	In vitro, RT-PCR, In situ hybridization, Immunocytochemistry.	To describe the response of PR to injury and hormone treatment.	PROG increases BDNF expression and protein in motoneurons in injured rats. These increases were correlated with increased TrkB and phosphorylated CREB in motoneurons.
6	Progesterone increases Brain-Derived Neurotrophic Factor Expression and Protects Against Glutamate Toxicity in a Mitogen-Activated Protein Kinase-and Phosphoinositide-3 Kinase-Dependent Manner in Cerebral Cortical Explants Paramjit Kaur et al., 2007. [101]	In vitro, ELISa RT-PCR.	To examine frontal and cingulate cerebral cortex explants from mice treated with PROG in vitro.	PROG induces a 75% increase in BDNF expression in explants of the cerebral cortex, with a nearly identical effect on BDNF protein levels.
7	Progesterone modulates brain-derived neurotrophic factor and choline acetyltransferase in degenerating Wobbler motoneurons Gonzalez Deniselle et al., 2007. [102]	Ex vivo, In situ hybridization, Immunofluorescence.	To examine steroid and BDNF expression and protein in the spinal cord and in muscle atrophy in wobbler rodents.	BDNF expression was found in neurons of steroid-naïve wobbler mice compared to controls. PROG treatment increased BDNF expression in wobblers compared to untreated but not in controls.
8	Progesterone pre-treatment enhances serotonin-stimulated BDNF gene expression in rat C6 glioma cells through production of 5α-reduced neurosteroids Morita and Her, 2008. [103]	In vitro, Rat C6 glioma cells, RT-PCR.	To investigate the role of neurotransmitters on glial cell metabolism and function in rat glioma cells in vitro.	BDNF expression levels in both non-treated and PROG-pre-treated glioma cells were similarly elevated by serotonin treatment with a concentration-dependent effect of serotonin on BDNF gene expression.
9	The differences in neuroprotective efficacy of progesterone and medroxyprogesterone acetate correlate with their effects on brain-derived neurotrophic factor expression Jodhka et al., 2009. [104]	In vitro, Western blot, RT-PCR.	To determine which type of PROG receptor mediates the neuroprotective effect of PROG on BDNF.	PROG induces an increase in the BDNF protein levels in cerebral cortical explants in a concentration-dependent manner. PROG regulates BDNF expression through the classical PROG receptor.
10	Progesterone, BDNF and Neuroprotection in the Injured CNS Coughlan, Gibson, and Murphy, 2009. [105]	In vitro, RT-PCR.	To investigate the neuroprotective mechanism of PROG and BDNF.	PROG had no effect on BDNF expression in granule neurons. No neuroprotective role for PROG on BDNF was observed.
11	Progesterone inhibits estrogen-mediated neuroprotection against excitotoxicity by down-regulating estrogen receptor-β. Aguirre et al., 2010. [97]	Ex vivo, Western blot, RT-PCR.	To examine PROG and estrogen treatment in cultured hippocampal slices on levels of ERα and ERβ and BDNF.	Estrogen elevated ERβ expression and protein levels and did not modify ERα expression, but it did increase ERα protein levels and BDNF expression levels in hippocampal cells. PROG reversed the estrogen-elicited increases in ERβ, ERα protein, and BDNF expression levels.
12	Progesterone treatment alters neurotrophin/proneurotrophin balance and receptor expression in rats with traumatic brain injury Cekic et al., 2012. [106]	Ex vivo, Western blot.	To characterize the expression of BDNF isoforms following PROG treatment for traumatic brain injury.	PROG reduces levels of pro-BDNF and TrkB post-brain injury. Mature BDNF was decreased at 24 and 72 h.
13	Progesterone increases the release of brain-derived neurotrophic factor from glia via progesterone receptor membrane component 1 (Pgrmc1)-dependent ERK5 signaling. Su et al., 2012. [107]	In vitro, RT-PCR, ELISA.	To study PROG-induced BDNF release and the extracellular signal-regulated kinase 5.	PROG and the membrane-impermeable PROG both induced BDNF release from glial cells and primary astrocytes, which lack the classical nuclear/intracellular PROG receptor but express membrane-associated PROG receptors.
14	Progesterone effects on neuronal brain-derived neurotrophic factor and glial cells during progression of Wobbler mouse neurodegeneration. Meyer et al., 2012. [108]	Ex vivo, In situ hybridization, Immunohistochemistry.	To compare PROG regulation of BDNF in motoneurons and oligodendrocytes of wobbler mice.	PROG upregulated low levels of BDNF expression in gray matter regions at the symptomatic stage of the disease and increased BDNF expression in late-stage wobblers. BDNF protein was normal in steroid-naive symptomatic wobblers.
15	Progesterone attenuates several hippocampal abnormalities of the wobbler mouse. Meyer et al., 2013. [109]	Ex vivo, In situ hybridization.	To examine the hippocampus of wobbler mice and their changes in response to PROG treatment.	Wobbler mice display decreased BDNF expression. PROG did not change the normal parameters in control mice and attenuated hippocampal abnormalities in wobblers.
16	Progesterone in the treatment of neonatal arterial ischemic stroke and acute seizures: Role of BDNF/TrkB signaling. Atif, Yousuf, and Stein, 2016. [110]	Ex vivo, Western blot.	To examine the effects of PROG on BDNF-TrkB signaling and inflammation following neonatal arterial ischemic stroke in mice.	PROG suppresses the expression of BDNF in mice with seizures at day 1, but at day 3, BDNF expression is comparable to controls. PROG treatment first inhibited TrkB expression at day 1 then increased TrkB receptor expression at day 3.
17	Progesterone modulates post-traumatic epileptogenesis through regulation of BDNF-TrkB signaling and cell survival-related pathways in the rat hippocampus. Ghadiri et al., 2019. [111]	Ex vivo, Western blot.	To study the effect of PROG on post-traumatic epileptogenesis survival-related pathways.	The duration of seizures was reduced in PROG-treated animals which showed an enhanced amount of BDNF in the ipsilateral hippocampus.
18	Progesterone’s Effects on Cognitive Performance of Male Mice Are Independent of Progestin Receptors but Relate to Increases in GABAA Activity in the Hippocampus and Cortex. Frye, Lembo, and Walf, 2021. [112]	Ex vivo, ELISA.	To evaluate the effect of PROG on the hippocampal and cortical levels of BDNF in mice.	PROG increased BDNF levels in the hippocampus, but not in the cortex, of male mice.

The modulatory effects of the combination of progesterone with estradiol on BDNF might be influenced by the dosage. Bimonte-Nelson et al. [98] showed that aged female rats receiving estradiol at a 1.5 mg/60-day regime exhibited increases in cortical levels of BDNF, and this was null when the animals also received progesterone at a 200 mg/60-day regime. In contrast, Saland, Schoepfer, and Kabbaj [113] reported that rats under a cyclic administration of estradiol and progesterone that received ketamine injections exhibited an increase in hippocampal BDNF levels that did not occur in those receiving only estradiol or progesterone.

Exposition time might be important for the neuroprotective effects of progesterone regulation on BDNF. Coughlan et al. [105] did not find an effect of progesterone on BDNF expression in the brain of mice who received a single progesterone dose of 8 mg/kg at the onset of a stroke; Cekic et al. [106] reported that levels of BDNF, pro-BDNF, and TrkB were reduced in the brains of rats treated with an 8mg/kg intraperitoneal injection of progesterone at 1 h, as well as subcutaneous injections at 6 and 24 h, continuing every 24 h post-injury. Alternatively, Yousuf et al. [57] showed that rats with strokes, who received progesterone (8 or 16 mg/kg) injections at 2 h, 6 h, and every 24 h until day 7 post-occlusion, restored BDNF levels at 3 and 7 days post-stroke. Similarly, Atif et al. [110] showed that mice with ischemic stroke that received 8 mg/kg progesterone injections at 1 h post-ligation, 3 h post-ligation, and every 24 h for 6 days exhibited a transient reduction in BDNF expression at day 1 that increased at day 3 through day 7. Finally, Ghadiri et al. [111] recently showed that male rats demonstrated an increase in hippocampal BDNF concentrations with low-dose progesterone treatment, while a high-dose progesterone treatment resulted in a decline in BDNF levels lower than those found in rats with brain injuries. It is possible that a time gap evidenced in BDNF response to progesterone occurs due to the demands of membrane-associated PRs. A critical range in progesterone dosage dictates whether BDNF expression will be induced by progesterone receptor activation or inhibited by CREB inhibition.

Jodhka et al. [104] identified that the progesterone regulation of BDNF expression in mice brains is mediated by the classical nuclear PR; Su et al. [107] showed that progesterone induces BDNF release in glial cells and astrocytes, which lack the classical nuclear PR, by activating membrane component 1 signaling (Pgrmc1). Nuclear steroid receptors stimulate gene expression by facilitating the assembly of basal transcription factors into a pre-initiation complex that requires additional, and sometimes common, coactivators. This means that the activation of ERs might impair PR-dependent gene expression via sequestering the common coactivator of CREB (CREB-binding protein) [114]. Competition between these nuclear receptors for limited concentrations of common coactivators is an integrative mechanism by which sex steroids concur in the genomic regulation of BDNF.

Although ERs are more abundantly present across brain tissues, the gene for BDNF contains an ERE sequence, allowing ER activation to more effectively regulate the neuroprotective effects of BDNF. The activation of non-nuclear receptors by progesterone influences *BDNF* expression and release in various neuronal and non-neuronal tissues and has been identified to positively regulate *BDNF* expression in spinal cord motor neurons, striatal neurons, and hippocampal and cortical neurons [99,101,102,108,109,112,115].

## 5. Androgen and BDNF Interactions

Several studies have elucidated that testosterone is able to increase the expression levels of BDNF in the high vocal center (HVC) of female and male birds, whereby it mediates the recruitment and survival of newborn neurons. This was determined in the studies by Xu et al. [116], Louissaint et al. [117], Fusani et al. [118], Hartog et al. [47], Ottem et al. [119], Li et al. [120], Fanaei et al. [121], and Falk Dittrich et al. [122]. Throughout life, new neurons arise from the ventricular zone of adult songbird brains and are recruited to the song control nucleus HVC, from which they extend projections to its target. This process of ongoing circuit integration is modulated by seasonal surges in systemic testosterone, supported by BDNF [123,124]. However, while it is known that singing upregulates HVC’s BDNF expression and that BDNF mediates androgen-induced HVC neuronal recruitment, the expression of BDNF is found to be diminished in aromatase-inhibited birds [125]. It is important to point out that brain aromatase converts circulating testosterone to estradiol, and thus, the HVC is exposed to both androgenic and estrogenic stimulation. Considering sexual dimorphism in BDNF expression in the canary HVC, where adult males exhibit higher levels than females, along with evidence of testosterone-induced increase in BDNF levels in adult female HVC, the indirect effect of testosterone on BDNF expression may occur either via its metabolite estradiol or testosterone-induced increases in singing activity. Concurrently, Allen et al.’s [126] experiments in adolescent male rats and macaques posited that neither gonadectomy nor testosterone replacement altered BDNF or TrkB expression levels in the hippocampal tissue of the animals. Together, these findings suggest that local changes in BDNF expression likely follow estradiol availability and, possibly, that gonadectomy or testosterone replacement did not alter the hippocampal production of estradiol.

**Table 3 ijms-26-02532-t003:** Data summary of interactions between testosterone and BDNF.

	Title	Design/Methods	Experiment	Main Findings
1	Brain-derived neurotrophic factor regulates expression of androgen receptors in perineal motoneurons Al-Shamma and Arnold, 1997. [127]	In vitro, Immunohistochemistry.	To examine steroid receptor expression in motoneurons of the SNB in mice.	Axonal transport disruption downregulates AR expression in motoneurons, and BDNF treatment reverses it.
2	Estrogen-inducible, sex-specific expression of brain-derived neurotrophic factor mRNA in a forebrain song control nucleus of the juvenile zebra finch Dittrich et al., 1999. [93]	Ex vivo, In situ hybridization.	To examine the expression of AR, BDNF, and TrkB in the HVC, neostriatum, and archistriatum in zebra finches.	BDNF expression is increased in the HVC of male, but not female, zebra finches. Estrogen and aromatase inhibition induce premature stimulation and inhibition of the increased patterns of BDNF expression, respectively, in juvenile males.
3	BDNF mediates the effects of testosterone on the survival of new neurons in an adult brain Rasika, Alvarez-Buylla, and Nottebohm, 1999. [123]	In vitro, Immunohistochemistry.	To examine BDNF responses to testosterone treatment in the HVC of male canaria.	Testosterone treatment increases BDNF levels in the HVC of adult canaria. BDNF antibodies block the testosterone-induced increase in new neurons.
4	BDNF regulation of androgen receptor expression in axotomized SNB motoneurons of adult male rats Yang and Arnold, 2000. [128]	In vitro, Autoradiography.	To examine BDNF effects on the axotomy-induced loss of AR expression in SNB motoneurons in rats.	The delayed application of BDNF to axotomized SNB motoneurons restored AR expression the intact levels.
5	Blockade of endogenous neurotrophic factors prevents the androgenic rescue of rat spinal motoneurons Xu et al., 2001. [116]	Ex vivo, Histology.	To exploit motoneuron cell death in the SNB of mice and the effect of androgen.	The blockage of TrkB activity prevented the androgenic sparing of SNB motoneurons. This did not reduce the SNB motoneuron number.
6	Coordinated interaction of neurogenesis and angiogenesis in the adult songbird brain Louissaint et al., 2002. [117]	In vitro, RT-PCR, Immunohistochemistry, ELISA.	To investigate testosterone-related angiogenesis and neuronal recruitment in adult songbird neostriatum.	HVC endothelial cells produce BDNF in a testosterone-dependent manner.
7	Aromatase inhibition affects testosterone-induced masculinization of song and the neural song system in female canaries Fusani et al., 2003. [118]	Ex vivo, In situ hybridization.	To investigate the role of estrogen in controlling the development of song structures (HVC) in female canaries.	The aromatase inhibition of testosterone-induced song motor development correlates with the inhibition of BDNF in HVC of adult female canaries and alters the song pattern.
8	Brain-Derived Neurotrophic Factor and Androgen Interact in the Maintenance of Dendritic Morphology in a Sexually Dimorphic Rat Spinal Nucleus Yang, Verhovshek, and Sengelaub, 2004. [129]	Ex vivo, Histochemistry.	To test BDNF and testosterone effects on dendritic morphology in motoneurons of the SNB in rats.	Testosterone or BDNF failed to support dendritic length or distribution. Treatment with testosterone plus BDNF restores dendritic morphology to the level of controls.
9	Androgen regulates trkB immunolabeling in spinal motoneurons Osborne, Verhovshek, and Sengelaub, 2007. [130]	Ex vivo, Immunohistochemistry.	To examine gonadal hormone regulation of BDNF systems in rodents’ spinal motoneurons.	TrkB receptor regulation is androgen-sensitive in motoneurons on the SNB. Castration-induced changes in SNB motoneurons are prevented by testosterone replacement.
10	Androgen-dependent regulation of brain-derived neurotrophic factor and tyrosine kinase B in the sexually dimorphic SNB Ottem et al., 2007. [131]	Ex vivo, In situ hybridization, RT-PCR.	To investigate the androgen regulation of BDNF protein in SNB motoneurons.	SNB motoneurons and the non-androgen-responsive motoneurons of the adjacent retrodorsolateral nucleus express BDNF and trkB. Testosterone regulates BDNF protein in SNB but not in the retrodorsolateral nucleus dendrites.
11	Differential expression and regulation of brain-derived neurotrophic factor mRNA isoforms in androgen-sensitive motoneurons of the rat lumbar spinal cord. Ottem et al., 2010. [119]	Ex vivo, In situ hybridization.	To examine the specific BDNF transcripts regulated by androgens in the SNB motoneurons of male rats.	BDNF isoforms containing exon VI were decreased in SNB motoneurons in an androgen-dependent manner but unaffected in retrodorsolateral motoneurons.
12	Androgen regulates brain-derived neurotrophic factor in spinal motoneurons and their target musculature. Verhovshek et al., 2010. [132]	Ex vivo, Immunohistochemistry, ELISA.	To examine the androgen regulation of BDNF in quadriceps and SNB motoneurons and their corresponding target musculature in male rats.	Castration reduced BDNF protein in the quadriceps, SNB motoneurons, and their target musculature, and this was prevented with testosterone replacement.
13	Modulatory Effects of Sex Steroid Hormones on Brain-Derived Neurotrophic Factor-Tyrosine Kinase B Expression during Adolescent Development in C57Bl/6 Mice. Hill et al., 2012. [133]	Ex vivo, Western blot.	To examine sex steroid hormones and neurotrophic signaling during adolescent development in a mouse model.	Castration and testosterone or DHT replacement had a receptor-dependent effect on BDNF-TrkB signaling in the forebrain and hippocampal regions of adolescent animals. Changes in BDNF-TrkB signaling in females did not align with changes in serum estrogen.
14	Androgen action at the target musculature regulates brain-derived neurotrophic factor protein in the SNB. Verhovshek and Sengelaub, 2013. [134]	Ex vivo, Immunohistochemistry.	To examine if testosterone regulates BDNF in SNB motoneurons of male rats by acting locally at the bulbocavernosus muscle.	Testosterone directly to the bulbocavernosus muscle maintains BDNF levels in SNB motoneurons intact after castration. AR blockage decreases BDNF compared with animals treated with intramuscular testosterone.
15	Regulatory mechanisms of testosterone-stimulated song in the sensorimotor nucleus HVC of female songbirds. Dittrich et al., 2014. [122]	Ex vivo, Microarray.	To examine the effects of testosterone on the anatomy and song control nucleus HVC of female European robins.	Testosterone induced differentiation, angiogenesis, and neuron projection morphogenesis. BDNF functions as a common mediator of the testosterone effects in HVC.
16	Testosterone enhances functional recovery after stroke through promotion of antioxidant defenses, BDNF levels and neurogenesis in male rats. Fanaei et al., 2014. [121]	Ex vivo, Immunohistochemistry.	To evaluate the effects of testosterone on BDNF and neurogenesis in a castrated male rat model of focal cerebral ischemia.	Testosterone increased BDNF levels and neurogenesis after focal cerebral ischemia.
17	The effect of adolescent testosterone on hippocampal BDNF and TrkB mRNA expression: Relationship with cell proliferation. Allen et al., 2015. [126]	Ex vivo, RT-PCR, In situ hybridization.	To examine the molecular mechanism underlying testosterone actions on postnatal neurogenesis and BDNF/TrkB levels in rhesus macaques and rats.	Gonadectomy or steroid replacement did not alter BDNF or TrkB expression levels in the hippocampus of young adult male rats or rhesus macaques. There was a positive correlation between cell proliferation and TrkB expression, but only when steroids were present.
18	Effects of testosterone on synaptic plasticity mediated by androgen receptors in male SAMP8 mice. Jia et al., 2016. [124]	Ex vivo, Western blot.	To study the protective role of testosterone on cognitive performance in an Alzheimer’s disease animal model.	The expression of BDNF and cyclic-AMP response element-binding protein (CREB)/CREB levels were elevated in testosterone-treated animals.
19	Hedonic sensitivity to low-dose ketamine is modulated by gonadal hormones in a sex-dependent manner. Saland, Schoepfer, and Kabbaj, 2016. [113]	Ex vivo, Western blot.	To investigate testosterone contribution in the rapid antidepressant-like effects of ketamine.	Testosterone treatment responsiveness was associated with higher hippocampal BDNF levels in female rats.
20	TrkB is necessary for male copulatory behavior in the Syrian Hamster (Mesocricetus auratus). Brague et al., 2018. [135]	Ex vivo, Western blot, RT-PCR.	To examine how TrkB and BDNF mediate testosterone effects on the medial preoptic nucleus in hamsters.	Testosterone treatment increased BDNF expression levels and, conversely, lowered the expression of TrkB receptors in the medial preoptic area of animals.
21	Prenatal Androgenization Induces Anxiety-Like Behavior in Female Rats, Associated with Reduction of Inhibitory Interneurons and Increased BDNF in Hippocampus and Cortex. Rankov Petrovic et al., 2019. [136]	Ex vivo, Immunohistochemistry, Western blot, ELISA.	To evaluate the influence of maternal hyperandrogenemia on offspring levels of BDNF in the hippocampus and cerebral cortex.	BDNF expression was increased in the hippocampus and cerebral cortex of prenatal hyperandrogenization offspring in comparison with the controls.
22	Deficiency in Androgen Receptor Aggravates the Depressive-Like Behaviors in Chronic Mild Stress Model of Depression. Hung et al., 2019. [137]	Ex vivo, Immunohistochemistry, Western blot, ELISA.	To exploit how AR and stress influence the onset of major depressive disorder.	Loss of AR affects depressive-like behaviors by modulating BDNF expression.
23	Effect of adolescent androgen manipulation on psychosis-like behaviour in adulthood in BDNF heterozygous and control mice. Du et al., 2019 [138]	Ex vivo, Western blot.	To examine how adolescent androgens influence psychosis-like behavior in adulthood and the role of BDNF in mice.	Testosterone and DHT treatment reduce the expression of dopamine transporter in the medial prefrontal cortex of mice. These effects are absent in BDNF heterozygous mice.
24	Dose-dependent effects of testosterone on spatial learning strategies and brain-derived neurotrophic factor in male rats. Zhang et al., 2020. [139]	Ex vivo, ELISA.	To investigate the effect of different doses of testosterone on spatial learning strategies in male rats.	Low testosterone doses increased total BDNF in the striatum, and high doses increased total BDNF in the hippocampus.

The interaction between BDNF and testosterone seems to be necessary to maintain dendritic morphology in SNB motoneurons. Testosterone can regulate the expression of AR and TrKB in spinal motoneurons [130], and BDNF signaling through TrkB is able to regulate the expression of androgen receptors (ARs) [127,128,138,140]. Furthermore, BDNF expression is shown to be upregulated by testosterone signaling in AR-expressing motoneurons of the vastus lateralis and the spinal nucleus of the bulbocavernosus and their corresponding muscles [132,134]. Together, these findings suggest a mechanistic interaction between BDNF and testosterone, whereby testosterone improves BDNF signaling and, consequently, its expression, which, in turn, improves testosterone sensitivity via AR regulation. In the experiment by Yang, Verhovshek, and Sengelaub [129], the treatment of castrated male rats with testosterone or BDNF alone was not sufficient to promote dendritic length and distribution in SNB motoneurons, whereas a combined treatment with testosterone and BDNF was able to restore the dendritic morphology of SNB. Additionally, data by Zhang et al. [139] indicate that different doses of testosterone may differently affect BDNF expression levels within the brain. In their experiments, young adult rats that received a low dose of testosterone (0.125 mg) showed increased BDNF concentrations in the striatum, and those who received a high dose (0.500 mg) had increased BDNF in their hippocampus.

The study by Hill et al. [133] reported region-specific and time-dependent sex differences in BDNF-TrkB expression and signaling during adolescence. They posited that serum testosterone levels increased in male mice from week 3 to 12, peaking at week 8 and then declining, and brain BDNF expression levels showed a positive correlation with serum testosterone, peaking from weeks 7 to 10. TrkB’s tendency to decrease when BDNF levels increase was noted, and castration did not affect BDNF expression in the mice brains. In females, BDNF expression did not change from week 3 to 12; however, TrkB activity was seen to increase from week 4, peak at week 6, and then decline. OVX resulted in an increase in BDNF expression levels that were not affected by estrogen replacement and a decrease in TrkB activity in these adolescent mice. In Brague et al.’s [135] experiment, testosterone also increased the expression of BDNF while reducing TrkB levels in the hypothalamus of male Syrian hamsters. Recently, a study by Rankov Petrovic et al. [136] revealed that adult female offspring from testosterone-treated pregnant female rats demonstrated increased BDNF levels in the hippocampus and cortex.

## 6. Summary and Conclusions

Sex steroids play an essential role in neuronal circuit formation and maintenance via the regulation of BDNF expression in neuronal proliferation, differentiation, and homeostasis at important areas of the brain, i.e., the cortex, hippocampus, hypothalamus, upper areas of the midbrain, and the cerebellum [18,25,28,31,38,55]. This occurs during the developmental stages throughout life.

The estrogen regulation of BDNF either via classical nuclear receptors or the activation of membrane-associated ERs and second messenger signaling pathways [141,142] is essential to neuronal function and is suggestibly a main regulator of TrkB expression. Whereas excessive concentrations of BDNF reportedly downregulate TrkB expression levels [143]. Further, BDNF autocrine and paracrine signal transduction regulates its own expression within neighbor cells [144], and this enhances ER activity, which, in turn, increases BDNF mRNA transcription in a synergistic manner [17,37,42,44,53,75,78,83,145]. This implies that the estrogen/ER system is able to regulate the BDNF/TrkB system; meanwhile, the inhibitory effect of the TrkB/PI3K/AKT pathway on ERE transcription indicates that the BDNF signal is important for limiting and controlling BDNF expression [50,53,64,78] (Figure 3).

A neuroprotective effect for the estrogen regulation of BDNF is consistently seen in individuals with various pathological conditions such as stress, hypertension, ischemic brain injury, and Parkinson’s and Alzheimer’s diseases [29,34,40,41,48,65,71,72,73,74,77,94,105,110,121,124]. A time- and region-specific influence found for estrogen on BDNF function in neuronal circuits likely depends on the presence and concentration of the sex steroid [34,47,116,117,118,119,120,123,129,130,132,134,136,139,146].

Although much less research is present on interactions between progesterone and BDNF, the direct regulation of BDNF expression was evidenced in cortical cells expressing the classical nuclear PR [40,104]. In addition, progesterone is able to increase BDNF synthesis and enhance neuroprotective cell responses in different ways [24,56,95,99,101,102,103,104,107,108,109,114]. For example, progesterone enhances anti-inflammatory processes and tissue recovery from neural injury [57,106,110,147,148,149,150].

Testosterone does not regulate the *BDNF* gene directly but rather via the activation of ARs, which translocate to the nucleus and commence gene expression at androgen-responsive elements sequences of androgen-responsive genes upstream of *BDNF* [122,127,128,151]. This regulation appears to be influenced by local factors. For instance, the findings of Zhang et al. [139] report that a low dose of testosterone increases BDNF levels in the striatum, while a high dose of testosterone increases BDNF levels in the hippocampus of male rats. Interestingly, the BDNF/TrkB system positively regulates the expression of ARs in neurons, potentiating testosterone regulation on the BDNF gene [129].

Regarding progesterone, studies have reported various effects of interactions between combinations of progesterone and estradiol on BDNF expression [49,84,96,98,99,102,112,113]. It is possible that their competitive activation of nuclear receptors might deplete co-activators, or their signaling through membrane-associated receptors might result in a gain or a loss of effect on BDNF regulation. Further studies on the mechanistic interactions between sex steroids in the regulation of gene expression are necessary to further understand progesterone and estradiol.

Sex steroids regulate the expression of BDNF either directly via nuclear receptors or by altering intracellular pathways that affect the genomic regulation of BDNF. This, at times, can represent concurrent mechanisms (Table 4).

The body of experimental research addressing steroid regulation of the neurotrophic factor BDNF supports the importance of steroid hormones in most essential processes of neuronal plasticity and functions. This synthesis will help expand our knowledge of the physiology of sex hormones in the brain and provide a broader comprehension of the events associated with hormone decline and therapy, including oncogenic risks.

## Figures and Tables

**Figure 1 ijms-26-02532-f001:**
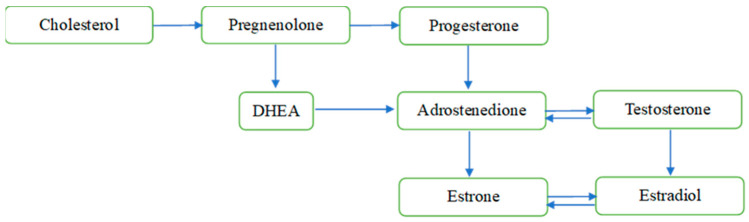
Overview of the de novo sex steroid synthesis within the mitochondria. Notes: The figure does not display all intermediate steroids, pathways, or enzymes. DHEA = dehydroepiandrosterone.

**Figure 2 ijms-26-02532-f002:**
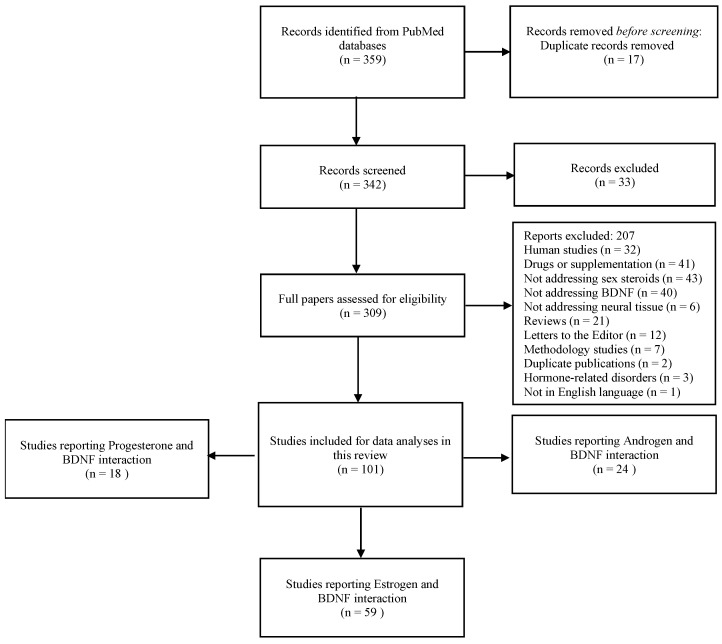
Search flowchart.

**Figure 3 ijms-26-02532-f003:**
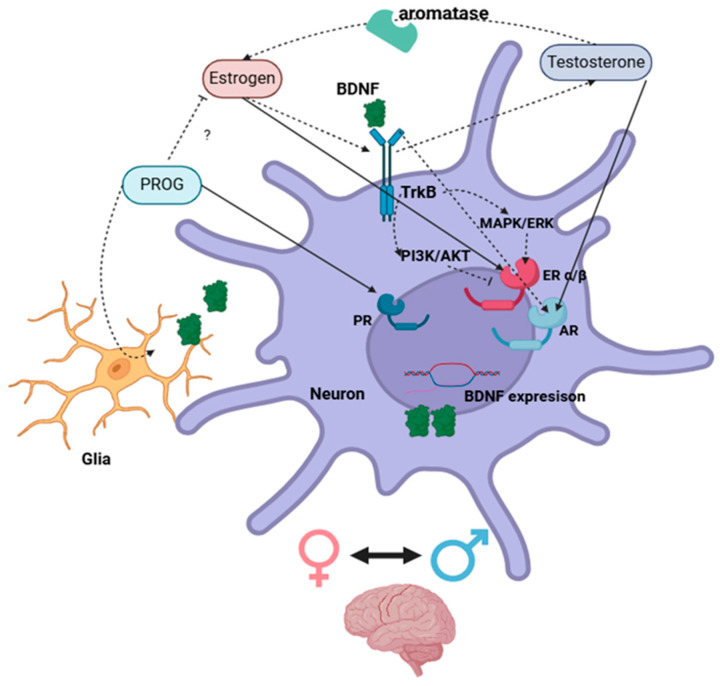
Schematic illustration of direct (complete arrow) and indirect (dotted arrow) interactions between sex steroids and BDNF systems via the activation of nuclear receptors in neuronal cells within the brain. ? = Unknown mechanisms. Created with https://www.biorender.com/.

**Table 1 ijms-26-02532-t001:** Data summary of interactions between estrogen and BDNF.

	Title	Design/Methods	Aim/Experiment	Main Findings
1	Interactions of estrogen with the neurotrophins and their receptors during neural development. Miranda, Sohrabji and Toran-Allerand, 1994. [17]	In vitro, Cell differentiation, P12 cells.	To examine interactions between estrogen and neurotrophins in cortical neurons.	Cortical neurons co-express BDNF, p75, and TrkB, and basal forebrain neurons only express neurotrophin receptors.
2	The effect of ovariectomy and estrogen replacement on brain-derived neurotrophic factor messenger ribonucleic acid expression in cortical and hippocampal brain regions of female Sprague Dawley rats. Singh, Meyer, and Simpkins, 1995. [18]	Ex vivo, mRNA analysis, In situ hybridization.	To investigate whether estradiol affects cholinergic function by modulating the levels of neurotrophic factors.	There was a reduction in BDNF levels in the rats’ frontal, parietal, and temporal cortices by 28 after OVX.
3	Levels of trkA and BDNF mRNA, but not NGF mRNA, fluctuate across the estrous cycle and increase in response to acute hormone replacement Gibbs, 1998. [24]	Ex vivo, mRNA analysis by In situ hybridization.	To examine the levels of BDNF in mice’s hippocampus and physiological changes in circulating sex steroids.	BDNF expression levels in CA1 and CA3/4 fluctuated across the estrous cycle. BDNF expression increased in the dentate granule cell layer, region CA1, and region CA3/4 in OVX animals 53 h after estrogen and 5 h after PROG treatment.
4	Brain-derived neurotrophic factor mediates estrogen-induced dendritic spine formation in hippocampal neurons Murphy, Cole, and Segal, 1998. [25]	In vitro, Cell differentiation, Cultured hippocampal neurons.	To examine estrogen and BDNF regulation of glutamic acid decarboxylase expression in hippocampal cultures.	Estrogen decreases BDNF expression in hippocampal neurons within 24 h, which suppresses inhibition and increases the excitatory tone, leading to an increase in dendritic spine density in pyramidal neurons.
5	Region- and peptide-specific regulation of the neurotrophins by estrogen Jezierski and Sohrabji, 2000. [26]	Ex vivo, Protein concentrations by Enzyme-Linked Immunosorbent Assay (ELISA) assay.	To examine the effect of estrogen on BDNF expression in the olfactory bulb and the cingulate cortex in OVX mice.	Estrogen regulation of BDNF is region-specific. It increases BDNF expression in mice olfactory bulbs and diagonal bands of Broca but decreases in the cingulate cortex.
6	Neurotrophin expression in the reproductively senescent forebrain is refractory to estrogen stimulation Jezierski and Sohrabji, 2001 [27]	Ex vivo, Protein concentrations, Western blot.	To compare the estrogen regulation of neurotrophin ligands and receptors in young adult and senescent mice diagonal bands of Broca.	Estrogen increases BDNF and TrKB expression in the olfactory bulb and horizontal limb and decreases p75NRT expression in young mice but increases it in senescent mice. Senescent mice have higher ERα expression but very low steroid receptor coactivator (SRC-1) expression in the olfactory bulb.
7	Expression and estrogen regulation of brain-derived neurotrophic factor gene and protein in the forebrain of female prairie voles Liu et al., 2001. [28]	Ex vivo, Protein immunoreactive staining and mRNA labeling.	To map BDNF immunoreactive staining and mRNA labeling throughout the forebrain in female prairie voles.	Estrogen-treated mice have higher levels of BDNF in the DG and CA3 regions of the hippocampus, as well as in the basolateral nucleus of the amygdala than controls.
8	Estrogen and exercise interact to regulate brain-derived neurotrophic factor mRNA and protein expression in the hippocampus Berchtold et al., 2001. [29]	Ex vivo, ELISA and in situ hybridization.	To investigate estrogen and exercise interaction in BDNF regulation.	Exercise increases hippocampal BDNF expression and protein levels in female mice, which is reduced in the absence of estrogen in a time-dependent manner.
9	Estrogen stimulates brain-derived neurotrophic factor expression in embryonic mouse midbrain neurons through a membrane-mediated and calcium-dependent mechanism Ivanova et al., 2001. [30]	Ex vivo and in vitro, ELISA, Cell culture and RT-PCR analyses.	To investigate if estrogen influences dopaminergic cell differentiation through a BDNF-dependent mechanism in the midbrains of mice.	Estrogen upregulates BDNF expression in the midbrains of mice and has a stimulatory effect on dopaminergic neuron differentiation by coordinating BDNF expression.
10	Estrogen Regulates the Development of Brain-Derived Neurotrophic Factor mRNA and Protein in the Rat Hippocampus Solum and Handa, 2002. [31]	Ex vivo, Immunocytochemistry, RT-PCR, Western blot analysis.	To examine gonadectomy and estrogen replacement effects on the BDNF system in mice’s developing hippocampus.	ERα and BDNF colocalize in cells within the developing hippocampus. BDNF expression levels reduce within 7 days in postnatal gonadectomized male rats, and estrogen treatment restores BDNF levels in intact animals. No changes were found in TrkB levels.
11	Estrogen enhances retrograde transport of Brain-Derived Neurotrophic Factor in the Rodent Forebrain Jezierski and Sohrabji, 2003. [32]	Ex vivo, Immunohisto-chemisthy.	To examine the effect of estrogen on the retrograde transport of BDNF in the diagonal band of Broca and its forebrain target in mice.	Estrogen-treated animals had greater numbers of neurons with retrogradely labeled BDNF than controls.
12	Influence of estradiol, stress, and 5-HT2A agonist treatment on brain-derived neurotrophic factor expression in female rats Cavus and Duman, 2003. [33]	Ex vivo, In situ hybridization.	To examine the estrous cycle and BDNF expression in the hippocampus and cortex of mice.	BDNF expression levels in the DG and the medial prefrontal cortex decrease when estradiol levels are highest. Acute estradiol treatment decreased hippocampal BDNF expression in acute OVX but did not affect chronic OVX animals.
13	Oestrogen regulates sympathetic neurite outgrowth by modulating brain derived neurotrophic factor synthesis and release by the rodent uterus Krizsan-Agbas et al., 2003. [34]	Ex vivo, ELISA, In situ hybridization.	To examine the role of neurotrophins and estrogen in uterine sympathetic nerve remodeling.	Estrogen increases BDNF expression and protein in the myometrium and endometrium of OVX mice.
14	Estrogen affects BDNF expression following chronic constriction nerve injury Zhao et al., 2003 [35]	Ex vivo, Radioimmunoassay, ELISA and RT-PCR.	To investigate the effect of estrogen on BDNF expression in neuropathic pain in a chronic constriction injury model of mice.	Mice with higher estrogen were more sensitive to thermal stimuli and had higher levels of BDNF expression and protein levels.
15	Anatomical evidence for transsynaptic influences of estrogen on brain-derived neurotrophic factor expression Blurton-Jones, Kuan, and Tuszynski, 2004. [36]	Ex vivo, Immunohistochemistry.	To examine the localization of estrogen receptors and BDNF in the brains of adult mice.	ERα and BDNF colocalize in the hypothalamus, amygdala, prelimbic cortex, and ventral hippocampus. ERβ and BDNF do not colocalize in any brain regions.
16	Environmental enrichment reduces the mnemonic and neural benefits of estrogen Gresack and Frick, 2004. [37]	Ex vivo, ELISA.	To observe environmental factors that influence mnemonic and neural response to estrogen in mice.	Estrogen decreased hippocampal BDNF in mice in standard conditions but not enriched ones.
17	Inhibition of tyrosine kinase receptor type B synthesis blocks axogenic effect of estrogen on rat hypothalamic neurones in vitro Brito, Carrer, and Cambiasso, 2004. [38]	In vitro, Western blot, Immunohistochemistry.	To examine the estrogen-induced axogenic response and upregulation of TrkB in neuronal and glial cultures of male mice.	An increase in TrkB is necessary for estrogen to exert its axogenic effect on male-derived neurons.
18	Effects of estrogen treatment on expression of brain-derived neurotrophic factor and cAMP response element-binding protein expression and phosphorylation in rat amygdaloid and hippocampal structures Zhou et al., 2005. [39]	Ex vivo, In situ PCR, Immunohistochemistry.	To examine the effect of estrogen on CREB expression, phosphorylation, and BDNF expression in the amygdala and hippocampus of mice.	Estrogen increased BDNF expression levels in mice’s amygdala, CA1, and CA3 regions of the hippocampus and increased pCREB in the medial and basomedial regions but not the central or basolateral amygdala.
19	Estradiol to aged female or male mice improves learning in inhibitory avoidance and water maze tasks Frye, Rhodes, and Dudek, 2005. [40]	Ex vivo, ELISA, Radioimmunoassay.	To evaluate the mnemonic effects of post-training estradiol in aged male mice.	BDNF levels decreased in the hippocampus of trained mice 1 h following estradiol exposure.
20	Sex and ovarian steroids modulate brain-derived neurotrophic factor (BDNF) protein levels in rat hippocampus under stressful and non-stressful conditions Franklin and Perrot-Sinal, 2006. [41]	Ex vivo, ELISA.	To investigate stress, sex hormones, and BDNF protein levels in CA1, CA3, and DG subregions of mice hippocampus.	Females have higher levels of BDNF in CA3 and lower levels in DG relative to males. Stress decreases BDNF in CA3 in all animals. Stress increases BDNF levels in the DG of PROG-treated OVX mice while decreasing in controls.
21	17β-estrogen Attenuates Hippocampal Neuronal Loss and Cognitive Dysfunction Induced By Chronic Restraint Stress in Ovariectomized Rats Takuma et al., 2007. [42]	Ex vivo, ELISA, RT-PCR.	To evaluate the effect of estrogen on cognitive function in rodents under stress environments.	OVX or chronic stress decreases the levels of hippocampal BDNF expression in the CA3 region. Estrogen attenuates the stress-induced decrease in hippocampal BDNF expression levels in OVX rats.
22	Mode of action and functional significance of estrogen-inducing dendritic growth, spinogenesis, and synaptogenesis in the developing Purkinje cell Sasahara et al., 2007. [43]	Ex vivo, RT-PCR Immunocytochemistry.	To analyze the expression of BDNF and estrogen in Purkinje cells of neonatal and cytochrome P450 aromatase knock-out rodents.	Estrogen induces the expression of BDNF in mouse cerebella and promotes dendritic growth of Purkinje cells during development.
23	17β-estrogen protects depletion of rat temporal cortex somatostatinergic system by β-amyloid Aguado-Llera et al., 2007. [44]	Ex vivo, ELISA RT-PCR Immunohistochemistry.	To investigate estrogen treatment in amyloid-beta-related changes in neuronal cells in the hippocampi of rodents.	Estrogen increased BDNF expression in hippocampal cells both in the absence or presence of estrogen.
24	Estrogen receptor β protects against acoustic trauma in mice Meltser et al., 2008. [45]	Ex vivo, Immunoassays, RT-PCR, Western blots.	To examine the role of ERs in response to auditory trauma.	BDNF expression was more pronounced in wild-type mice compared to ER-deficient mice.
25	β-estrogen induces synaptogenesis in the hippocampus by enhancing brain-derived neurotrophic factor release from dentate gyrus granule cells Sato et al., 2007. [46]	In vitro, hippocampal neuron cultures.	To examine the effect of estrogen on synaptogenesis in hippocampal neuronal cell cultures.	The effects of estrogen in the hippocampal and subregional hippocampal neurons were independent of nuclear ERs and dependent on BDNF. Estrogen enhanced BDNF release from DG granule cells via nuclear ER-independent and PKA-dependent mechanisms.
26	Brain-derived neurotrophic factor signaling in the HVC is required for testosterone-induced song of female canaries Hartog et al., 2009 [47]	Ex vivo, In situ transfection, Western blot.	To examine BDNF and T4-dependent development of the song system in female canaries.	The testosterone-induced song system is blocked by concurrent inhibition of the vascular endothelial growth factor receptor tyrosine kinase, which is reversed by BDNF.
27	Involvement of Brain-Derived Neurotrophic Factor and Neurogenesis in Oestrogen Neuroprotection of the Hippocampus of Hypertensive Rats. Pietranera et al., 2010. [48]	Ex vivo, ELISA. Immunocytochemistry.	To evaluate estrogen treatment in hypertensive rats and BDNF expression.	Hypertensive rats exhibit decreased expression and protein levels of BDNF in the DG without changes in CA1 or CA3 pyramidal cell layers. Estrogen increases BDNF expression in the DG and BDNF protein in the whole hippocampus.
28	Estrogen reduces BDNF level, but maintains dopaminergic cell density in the striatum of MPTP mouse model. Tripanichkul, Gerdprasert, and Jaroensuppaperch, 2010. [49]	Ex vivo, Immunohistochemistry.	To examine the effects of estrogen treatment on BDNF expression and the density of DA neurons in the striatum of MPTP mice.	Estrogen impaired dopaminergic denervation and decreased the striatal BDNF upregulation triggered by MPTP.
29	Full length TrkB potentiates estrogen receptor alpha mediated transcription suggesting convergence of susceptibility pathways in schizophrenia. Wong, Woon, and Weickert, 2011. [50]	In vitro, Cull culture, Immunofluorescence, Western blot.	To examine ERα interaction with TrkB in neuronal and non-neuronal cell lines.	TrkB activation increases transcription at EREs, independent of exogenous estrogen, and further potentiates the effect of estrogen-ERα-mediated transcription.
30	17β-Estrogen replacement in young, adult and middle-aged female ovariectomized rats promotes improvement of spatial reference memory and an antidepressant effect and alters monoamines and BDNF levels in memory- and depression-related brain areas. Kiss et al., 2012 [51]	Ex vivo, ELISA.	To evaluate the effects of chronic treatment with estrogen on cognition and depressive-like behaviors in young, adult, and middle-aged female rats.	Both young mice and estrogen-treated OVX mice have higher BDNF levels. The young estrogen-treated OVX group presented higher BDNF levels compared to adult and middle-aged estrogen-treated animals.
31	17β-estrogen Regulates the Sexually Dimorphic Expression of BDNF and TrkB Proteins in the Song System of Juvenile Zebra Finches. Tang and Wade, 2012. [52]	Ex vivo, Western blot.	To examine BDNF isoforms and TrkB expression in the developing song system of juvenile males and females zebra finch treated with estrogen.	Estrogen modulates BDNF expression and TrkB in the song system of juveniles of both sexes.
32	Estradiol acts via estrogen receptors alpha and beta on pathways important for synaptic plasticity in the mouse hippocampal formation. Spencer-Segal et al., 2012. [53]	Ex vivo, In situ hybridization, Western blot.	To study estradiol systems and pathways related to plasticity and learning.	Estradiol increased phosphorylated Akt phosphorylated TrkB receptor in the hippocampus. These effects were abolished in ERα and ERβ knockout mice.
33	Estrone is neuroprotective in rats after traumatic brain injury. Gatson et al., 2012. [54]	Ex vivo, Immunohistochemistry, Western blot.	To study the role of estrone in traumatic brain injury in rats.	Cortical levels of phospho-ERK1/2 are increased by estrone, which was associated with an increase in phospho-CREB levels and BDNF expression.
34	Estradiol promotes purkinje dendritic growth, spinogenesis, and synaptogenesis during neonatal life by inducing the expression of BDNF. Haraguchi et al., 2012. [55]	Ex vivo, Immunohistochemistry.	To study estradiol and cerebellar neuronal circuit formation, dendritic growth, spinogenesis, and synaptogenesis in the Purkinje cell of wild vs. aromatase KO mice.	Estradiol increased neuroplasticity in all Purkinje cells. ER antagonist decreases BDNF levels in all mice. BDNF administration to ER antagonist-treated mice increased Purkinje dendritic growth.
35	Central expression and anorectic effect of brain-derived neurotrophic factor are regulated by circulating estrogen levels. Zhu et al., 2013. [56]	Ex vivo, RT-PCR.	To study if estradiol modulates the anorectic effect of BDNF in OVX rats.	BDNF expression is elevated in the hypothalamus during oestrus, following the estradiol peak, and after estradiol treatment.
36	Post-stroke infections exacerbate ischemic brain injury in middle-aged rats: Immunomodulation and neuroprotection by PROG. Yousuf et al., 2013 [57]	Ex vivo, ELISA.	To evaluate the effect of systemic inflammation on stroke outcomes and PROG neuroprotection in middle-aged rats.	Serum BDNF levels decrease in systemic inflammation conditions. PROG decreases cytokine levels and systemic inflammation and restores BDNF levels within 3 and 7 days post-stroke.
37	Long-term OVX increases BDNF gene methylation status in mouse hippocampus. Moreno-Piovano et al., 2014. [58]	Ex vivo, RT-PCR, Immunohistochemistry. Real-time quantitative methylation-specific PCR.	To determine if the post-OVX timeframe elapsed before E treatment is critical for the estrogen induction of neurotrophins BDNF in the rodents’ hippocampus.	Early estrogen-treated animals showed increased BDNF expression and a higher activity of BDNF II, IV, and V promoters. Late treated animals did not show estrogen induction of neurotrophins, and the methylation levels of the regulatory sequences of the BDNF gene were higher than in the early-treated animals.
38	17α-Oestrogen-Induced Neuroprotection in the Brain of Spontaneously Hypertensive Rats. Pietranera et al., 2014. [59]	Ex vivo, Immunohistochemistry, In situ hybridization.	To investigate estrogen and pathological changes in the hippocampus and hypothalamus of hypertensive rats.	Estrogen treatment enhanced the number of cells and increased BDNF expression in the CA1 region and DG.
39	17β-Estrogen regulates histone alterations associated with memory consolidation and increases Bdnf promoter acetylation in middle-aged female mice. Fortress et al., 2014 [60]	Ex vivo, RT-PCR, Western blot.	To examine the effects of estrogen infusion in the mice hippocampal on object recognition and spatial memory.	Estrogen specifically increased acetylation at BDNF promoters pII and pIV in the dorsal hippocampus of young and middle-aged mice despite age-related decreases.
40	Chronic estrogen treatment decreases Brain Derived Neurotrophic Factor (BDNF) expression and monoamine levels in the amygdala—Implications for behavioral disorders. Balasubramanian et al., 2014. [61]	Ex vivo, Immunoassay, PCR.	To verify whether chronic low estrogen doses cause anxiety-like disorder by altering BDNF and monoamine levels in rats’ hippocampus and amygdala.	Chronic estrogen treatment decreased BDNF expression and protein levels in the central amygdala, which was accompanied by a reduction in dopamine levels. No changes were observed in the hippocampus.
41	Analyzing the influence of BDNF heterozygosity on spatial memory response to 17β-estrogen. Wu et al., 2015. [62]	Ex vivo, Immunofluorescence Western blot.	To test if disruption to the estrogen–parvalbumin pathway alters learning and memory and BDNF levels in mice.	Estrogen replacement prevented the reduction in BDNF and parvalbumin protein levels in the dorsal hippocampus and CA1. BDNF heterozygote mice showed either no response or an opposite response to estrogen treatment.
42	Aging-induced changes in sex-steroidogenic enzymes and sex-steroid receptors in the cortex, hypothalamus and cerebellum. Munetomo et al., 2015. [63]	Ex vivo, RT-PCR.	To examine age-induced changes in sex-steroidogenic enzymes and sex-steroid receptors in 3-, 12-, and 24-month-old male rats’ cerebral cortex, hypothalamus, and cerebellum.	BDNF expression decreased from 3 to 24 m in the cerebral cortex but increased in the hypothalamus and did not change in the cerebellum. The expression levels of AR, ERα, and ERβ were higher in the Hypothalamus than in the cerebral cortex and cerebellum.
43	ERα Signaling Is Required for TrkB-Mediated Hippocampal Neuroprotection in Female Neonatal Mice after Hypoxic–Ischemic Encephalopathy. Cikla et al., 2016. [64]	Ex vivo, Western blot, RT-PCR.	To investigate how hypoxia induces ERα expression in the female neonatal hippocampus.	TrkB phosphorylation post-trauma is greater in females than males after selective TrkB agonist therapy and depends on the presence of ERα. TrkB agonist therapy decreases c-caspase-3, but only in the presence of ERα.
44	Combined exercise ameliorates OVX-induced cognitive impairment by enhancing cell proliferation and suppressing apoptosis. Kim et al., 2016. [65]	Ex vivo, Immunohistochemistry, Western blot.	To evaluate the effects of exercise on memory deficits, cell proliferation, and apoptosis in the hippocampus of OVX rats.	The expression of BDNF and TrkB decreased in the DG, together with memory decrease in OVX rats. These expression levels increased in the exercise group.
45	Selective Oestrogen Receptor Agonists Rescued Hippocampus Parameters in Male Spontaneously Hypertensive Rats. Pietranera et al., 2016. [66]	Ex vivo, Immunocytochemistry, RT-PCR.	To examine which type of ER is involved in low BDNF expression in the hippocampus of hypertensive rats.	ERα agonist slightly increased *BDNF* expression but had no effect on the number of doublecortin progenitors. Treatment with ERβ agonist increased *BDNF* expression and doublecortin progenitors.
46	Regulation of endometrial cell proliferation by estrogen-induced BDNF signaling pathway. Dong et al., 2017. [67]	In vitro, Human endometrial cancer cells, Western blot.	To investigate the role of the *BDNF* Val66Met polymorphism in regulating proliferation in endometrial cells treated with estrogen.	BDNF signaling pathway activates with estrogen stimulation. BDNF production is induced by estrogen, and the *BDNF* Val66Met is a loss-of-function polymorphism in the regulation of endometrial cell proliferation.
47	Sex differences in the effect of chronic mild stress on mouse prefrontal cortical BDNF levels: A role of major ovarian hormones. Karisetty et al., 2017. [68]	Ex vivo, Westerns blot, RT-PCR.	To examine the effect of sex hormones on depression-like phenotypes in mice exposed to 21-day chronic stress.	There was a decrease in the BDNF protein levels in intact females but not in OVX or male mice. Estrogen treatment, and not PROG, increased BDNF expression in the prefrontal cortex of the stressed mice with OVX.
48	Sex differences and estrogen regulation of BDNF gene expression, but not propeptide content, in the developing hippocampus. Kight and McCarthy, 2017. [69]	Ex vivo, RT-PCR, ELISA, Western blot.	To examine the downstream effects of estrogen on hippocampal cell proliferation and BDNF expression in male and female neonatal rats.	Estrogen had opposite effects on BDNF expression in different areas of the neonatal hippocampus. In the CA1, BDNF increased but decreased in DG. Blocking estrogen signaling decreased BDNF expression in the DG in males but not females. No differences were observed in pro-BDNF protein.
49	Estrogen receptor β deficiency impairs BDNF–5-HT2A signaling in the hippocampus of female brain: A possible mechanism for menopausal depression. Chhibber et al., 2017. [70]	Ex vivo, In vitro, Western blot.	To examine the ER subtypes in the regulation of BDNF and serotonin signaling in mice.	BDNF was downregulated in ERβ−/− mice in a brain-region-specific manner. There was a reduction in BDNF protein levels in the hippocampus of ERβ−/− mice and no changes in the cortex and hypothalamus. ERβ agonism enhanced BDNF/TrkB signaling pathways.
50	Additive antidepressant-like effects of fasting with β-Estrogen in mice. Wang et al., 2019. [71]	Ex vivo, Immunohistochemistry, Western blot.	To evaluate the antidepressant-like effects of acute fasting and estrogen treatment.	BDNF-TrkB signaling pathway was increased in the prefrontal cortex and hippocampus. Serum ghrelin and estrogen were increased by fasting plus estrogen.
51	Voluntary exercise and estrogen reverse OVX-induced spatial learning and memory deficits and reduction in hippocampal brain-derived neurotrophic factor in rats. Rashidy-Pour et al., 2019. [72]	Ex vivo, ELISA.	To investigate voluntary exercise and estrogen replacement in learning and memory deficits and hippocampal BDNF levels in OVX mice.	Either exercise and estrogen alone or their combination recovery the negative effects of OVX on learning and memory performance. Combined treatment does not potentiate the effect of either treatment alone.
52	Sex-dependent aberrant prefrontal cortex development in the adolescent offspring rats exposed to variable prenatal stress. Niu et al., 2020. [73]	Ex vivo, ELISA.	To examine neurochemical and behavioral changes in the offspring rats (adolescents) from rats treated with prenatal stress during the third week of gestation.	Prenatal stress increases BDNF levels in the prefrontal cortex of adolescent females. ER concentration increased with age in all animals.
53	Rapid effects of estrogen and its receptor agonists on object recognition and object placement in adult male zebrafish. Naderi et al., 2020. [74]	Ex vivo, Whole-brain RT-PCR.	To evaluate the effects of post-training estrogen in the consolidation of object recognition and object placement memory in adult male zebrafish.	Post-training estrogen treatment and selective ER agonist induced upregulation in the *BDNF* expression levels. BDNF expression increases with a high estrogen dose.
54	Reducing luteinizing hormone levels after OVX improves spatial memory: Possible role of brain-derived neurotrophic factor. Bohm-Levine et al., 2020. [75]	Ex vivo, Immunohistochemistry, Western blot.	To evaluate whether lowering luteinizing hormone increases BDNF expression levels.	Gonadotropin-releasing hormone receptor antagonists lower luteinizing hormone levels, and estrogen enhances spatial memory in OVX females, all of which are ineffective in the absence of TrKB. Both hormones increase BDNF expression in the hippocampus.
55	ERRγ ligand HPB2 upregulates BDNF-TrkB and enhances dopaminergic neuronal phenotype. Kim et al., 2021. [76]	In vitro, Immunohistochemistry, RT-PCR, Western blot.	To examine whether the ERRγ ligand regulates BDNF signaling and subsequent DAergic neuronal phenotype.	The ERRγ agonist increases BDNF expression and protein levels and TrkB expression in human neuroblastoma, differentiated Lund mesencephalic cells, and primary ventral mesencephalic neurons. Activation of ERK and phosphorylation of CREB induced BDNF upregulation in human neuroblastoma cells.
56	Previous oestradiol treatment during midlife maintains transcriptional regulation of memory-related proteins by ERα in the hippocampus in a rat model of menopause. Baumgartner et al., 2021. [77]	Ex vivo, RT-PCR, Western blot.	To examine whether previous estradiol treatment increases the levels of nuclear ERα, resulting in transcriptional regulation of proteins.	Continuous or previous estradiol treatments increase gene expression of BDNF.
57	17β-Estrogen activation of dorsal hippocampal TrkB is independent of increased mature BDNF expression and is required for enhanced memory consolidation in female mice. Gross et al., 2021. [78]	Ex vivo, RT-PCR, Western blot.	To examine the effects of hippocampal TrkB signaling on estrogen-induced enhancement of memory consolidation in object placement and recognition tasks.	Dorsal hippocampal estrogen infusion increased levels of phospho-TrkB and BDNF. The estrogen-induced increase in hippocampal BDNF expression was not required for hippocampal TrkB activation and was not inhibited by TrkB antagonism.
58	ERRγ Ligand Regulates Adult Neurogenesis and Depression-like Behavior in a LRRK2-G2019S-associated Young Female Mouse Model of Parkinson’s Disease [79]	Ex vivo, Immunohistochemistry, RT-PRC Western blot.	To investigate whether the ERRγ ligand HPB2 could be a novel therapeutic agent for regulating depressive behavior in a PD mouse model.	HPB2 upregulated BDNF/TrkB signaling in the DG of young female LRRK2-G2019S mice.
59	Activation of GPER1 by G1 prevents PTSD-like behaviors in mice: Illustrating the mechanisms from BDNF/TrkB to mitochondria and synaptic connection [80]	Ex vivo Immunohistochemistry, Western blot.	To investigate the mechanism of G1 to improve PTSD from brain-derived neurotrophic factor (BDNF)/tyrosine kinase receptor B (TrkB) signaling.	G15 (GPER1 inhibitor) and ANA-12 (TrkB inhibitor) blocked the ameliorative effects of G1 on PTSD-like behaviors and the aberrant expression of hippocampal synaptic and mitochondrial proteins in mice.

**Table 4 ijms-26-02532-t004:** Summarized highlights of findings from different studies.

Main Findings	Studies
BDNF gene expression is regulated via ERE and by the activation of nuclear steroid receptors, while the activation of some membrane-associate steroid receptors results in BDNF inhibition.	Frye, Rhodes, and Dudek 2005; Jodhka et al., 2009. [40,104]
While both BDNF and estrogen enhance BDNF expression and estrogen receptor activation, only the TrkB signal is able to inhibit ERE transcription and limit BDNF expression.	Cikla et al., 2016; Gross et al., 2021; Spencer-Segal et al., 2012; Wong, Woon, and Weickert 2011. [50,53,64,78]
Different sex steroid concentrations and combinations differently influence BDNF regulation, possibly due to the interactions of intracellular pathways.	Aguirre et al., 2010; Atif, Yousuf, and Stein 2016; Cekic et al., 2012; Coughlan, Gibson, and Murphy 2009. [97,105,106,110]
BDNF/TrkB signaling regulates the expression of androgen receptors.	Al-Shamma and Arnold 1997; Du et al., 2019; Halievski et al., 2015; Yang and Arnold 2000. [127,128,138,140]

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
