# Peer review of "Sex Steroids and Brain-Derived Neurotrophic Factor Interactions in the Nervous System: A Comprehensive Review of Scientific Data"

_ijms, 2025, doi:10.3390/ijms26062532_

Round 1

Reviewer 1 Report

Comments and Suggestions for Authors

This review written by De Assis et al., focuses on the impact of sex steroids on the nervous system.  Overall, the authors emphasize on the regulatory effect of estrogen and androgens on the brain-derived neurotrophic factor (BDNF) and its receptor. Besides gender dimorphism, direct and indirect regulatory mechanisms have been evaluated based on published preclinical investigations. Overall, the manuscript is well-written and put and conclusions are put into perspective. However, two major limitations need to be addressed:

1- While both estrogen and estrogen receptors are discussed, no emphasis was put on receptor-specific effects. This needs to be added to the manuscript as ERbeta and ERalpha may play distinct functions in the nervous system.

2- A summary table could describe key studies with their associated conclusions. 

Author Response

Reviewer 1: This review written by De Assis et al., focuses on the impact of sex steroids on the nervous system.  Overall, the authors emphasize on the regulatory effect of estrogen and androgens on the brain-derived neurotrophic factor (BDNF) and its receptor. Besides gender dimorphism, direct and indirect regulatory mechanisms have been evaluated based on published preclinical investigations. Overall, the manuscript is well-written and put and conclusions are put into perspective. 

Response: Thank you very much for reviewing our manuscript. Please find the detailed responses to your appointments below and the corresponding revisions/corrections highlighted in the re-submitted files.

Comments 1: While estrogen and estrogen receptors are discussed, no emphasis was placed on receptor-specific effects. This needs to be added to the manuscript as ERbeta and ERalpha may play distinct functions in the nervous system.

Response 1: Thank you for pointing this out. I agree with this comment. ERa and b are present in different distribution patterns throughout tissues and seem to exert different functions in different environments/contexts, this might be extended/include their role in BDNF regulation. Therefore, we have made insertions and adjustments to make it more clear that the regulatory role of estrogens on BDNF might be influenced by differences in the activity of estrogen receptors. Please, revisit the text between lines [177-195].

Comments 2: A summary table could describe key studies with their associated conclusions. 

Response 2: We found this idea very interesting and we understand that the experimental studies contributed with very specific pieces of the knowledge synthesized in this qualitative review. So we have added a new table (Table IV) to describe some appointments summarized from studies data. Please find it at the end of the document, line [899].

Reviewer 2 Report

Comments and Suggestions for Authors

Thank you very much for inviting me to contribute to this very interesting literature review on the effects of the interaction of sex steroids and brain-derived neurotrophic factor in the nervous system. The assigned topic seems to be extremely instony for understanding and expanding our knowledge of the physiology of sex hormones on the brain. In the future, this knowledge may have important direct clinical implications. 

Nevertheless, there are a few details that I believe should be improved:

First and foremost, there is no description of the methodology (other than a brief mention in the abstract). I suggest adding a separate section where the exact method of the literature review would be described, for example, using the PRISMA scheme. Describe what databases were searched and how. How many publications were used to perform the review, etc....  

I propose to add additional columns to the table placed on the publication, where it would state: 1. date of publication; 2. type of study: in vitro , in vivo or ex vivo. This would add clarity and make it easier to interpret the data. 

In summary and conclusion, it is worth adding why such studies are important and what relevance they can have for the patient in clinical management.

Author Response

Reviewer 2

Thank you very much for inviting me to contribute to this very interesting literature review on the effects of the interaction of sex steroids and brain-derived neurotrophic factors in the nervous system. The assigned topic seems to be extremely instony for understanding and expanding our knowledge of the physiology of sex hormones in the brain. In the future, this knowledge may have important direct clinical implications.

Response: We thank you for your time and attention, and we appreciate the meaningful insights.

Comments 1: First and foremost, there is no description of the methodology (other than a brief mention in the abstract). I suggest adding a separate section where the exact method of the literature review would be described, for example, using the PRISMA scheme. Describe what databases were searched and how. How many publications were used to perform the review, etc....

Response 1: We have added a Methods section and described it according to PRISMA recommendations. Also, we have performed a search update and included new papers that met our inclusion criteria. Please find it in between lines [107-129]

Comments 2: I propose to add additional columns to the table placed on the publication, where it would state: 1. date of publication; 2. type of study: in vitro , in vivo or ex vivo. This would add clarity and make it easier to interpret the data.

Response 2: Agree. We have, accordingly, included a new column containing the type of study and analyses; and we placed the date below the studies’ titles in the first column. Please, revisit the Tables. Please note that two new studies were added after a literature search update.

Comments 3: In summary and conclusion, it is worth adding why such studies are important and what relevance they can have for the patient in clinical management.

Response 3: We agree. We have added a closing paragraph expressing how the knowledge from this synthesis may help to understand clinical demands regarding the use and risks of hormone therapy for the brain’s health and give insights for new investigations. Please visit lines [408-413]

Round 2

Reviewer 2 Report

Comments and Suggestions for Authors

The manuscript in the revised version meets my expectations. The authors have met all the previous comments and made appropriate adjustments.
Thank you for inviting me to review this review.